# “Maybe it’s Not Just the Food?” A Food and Mood Focus Group Study

**DOI:** 10.3390/ijerph20032011

**Published:** 2023-01-21

**Authors:** Megan F. Lee, Douglas Angus, Hayley Walsh, Sally Sargeant

**Affiliations:** 1Faculty of Society and Design, Bond University, Gold Coast 4226, Australia; 2Faculty of Health, Southern Cross University, Gold Coast 4225, Australia

**Keywords:** food, mood, depression, mental health, psychological wellbeing, qualitative research, focus group, thematic template analysis

## Abstract

Epidemiological and intervention studies in nutritional psychiatry suggest that the risk of mood disorders is associated with what we eat. However, few studies use a person-centred approach to explore the food and mood relationship. In this qualitative study of 50 Australian participants, we explored individuals’ experiences with food and mood as revealed during focus group discussions. Using a thematic template analysis, we identified three themes in the food and mood relationship: (i) *social context: familial and cultural influences of food and mood*, (ii) *social economics: time, finance, and food security*, and (iii) *food nostalgia: unlocking memories that impact mood*. Participants suggested that nutrients, food components or food patterns may not be the only way that food impacts mood. Rather, they described the social context of who, with, and where food is eaten, and that time, finances, and access to healthy fresh foods and bittersweet memories of foods shared with loved ones all impacted their mood. Findings suggest that quantitative studies examining the links between diet and mood should look beyond nutritional factors and give increased attention to the cultural, social, economic, and identity aspects of diet.

## 1. Maybe it’s Not Just the Food? A Food and Mood Focus Group Study

Over 350 million people annually experience symptoms of mood disorders such as depression [1]. Although the relationship between nutritional intake and physiological health is well-studied [2,3], evidence of the impact of nutrition on mood is a new and growing field [4,5]. Nutritional psychiatry explores the relationship between nutrition and mental health [6,7]. Research suggests that high-quality diets rich in nutrient-dense vegetables, fruits, seeds, nuts, whole grains, and legumes [8] and reduced intakes of refined, sugary, and ultraprocessed foods [9] link to reductions in depressive symptoms [10]. Clinical trials demonstrating an inverse relationship between diet quality and symptoms of depression [11,12,13,14] indicate changes from an unhealthy to a healthy dietary pattern could form the basis of successful interventions for the treatment of depression. Systematic reviews and meta-analyses support these clinical findings [10,15,16], but the evidence is uncertain due to a large reliance on observational studies, high levels of unexplained heterogeneity, and risk of bias inherent in nutrition-related research [8,17].

### 1.1. Food, Mood, and Meaning

Although several qualitative studies have explored food and mood relationships, these focused on consumer preferences, rather than impacts on individuals’ mental health [18,19]. For example, a focus group study with 31 participants in the United Kingdom explored links between consumers’ food choices and their emotional states and suggested that high-quality food choices were due to increased consumer mood, while low-quality food choices were linked to habits learned from early childhood rewards [19]. In an unpublished master’s thesis, Houwen [18] interviewed 12 participants on the impact of mood on consumer product choices and found this hard to explore, as participants believed the food and mood relationship to be subconscious. A third qualitative study of 22 women in the United Kingdom that incorporated both focus groups and interviews concluded that food choices influenced mood, but that mood also influenced food choices [20]. The authors noted that participants described experiencing lower mood when consuming unhealthier foods, food addiction, and overeating, and reported experiencing greater mood when eating healthier foods and in a social context. However, there is no research that explores how “meaning” [21] and social context are attached to food choices and how this might impact mood/mental health.

A phenomenological viewpoint may provide broader insight on social contexts within which the food and mood relationships exist [22]. Phenomenology addresses intricacies of how individuals experience certain phenomena from their personal interpretations, perspectives, and contexts [23]. Phenomenological research sets aside the researchers preconceived ideas, biases, and experiences to fully engage with participant views [24], and phenomenological inquiry is relevant to investigating food–mood relationships from participants’ perspectives [25].

### 1.2. The Current Study

The concepts of healthy/unhealthy eating are shaped by individuals’ unique perceptions and meanings assigned to food choices [26]. Likewise, an individual’s dietary pattern is shaped by several influencing factors: life experiences, social contexts, beliefs, values, and expectations [27]. The impact of these different contexts means that a phenomenological exploration of individuals’ experiences is valuable in understanding how meaning is applied to food choices and their relation to mood/mental health. Here, we use data collected during focus group discussions and a person-centred, thematic analysis approach to explore these issues.

## 2. Materials and Methods

Nine focus groups, seven face-to-face and two online via Zoom, were conducted to explore individuals’ experiences with food choices and mood/mental health using the Consolidated criteria for Reporting Qualitative research (COREQ) [28]. See the Appendix A for the COREQ checklist [28]. This paper reports a second round of thematic findings focusing on social context from these focus groups. The first round of thematic findings focused on the meaning of food choices from these focus groups are reported in another paper by Lee, Bradbury, Yoxall, and Sargeant [26].

### 2.1. Participants and Procedure

Participants (N = 50) from Australian regional locations attended one of nine 60 min discussions. Excluding gender and age range—which were revealed during the group discussions—participant demographic information was not collected to facilitate comfort and promote open discussion. Participants consisted of 36 females (72%) and 14 males (28%). During the discussion, ages were revealed to be between 18 and 72 years of age, and all participants identified as either holding a current academic qualification or working towards a university degree. Participants were provided refreshments but no further compensation for participation. At this stage, participants self-disclosed dietary requirements for the refreshments. Most participants identified as being vegan, vegetarian, gluten-free, pescatarian, ketogenic or dairy-free. A total of 83 participants were recruited by email invitations circulated throughout a multicampus Australian university, or by social media advertisements. Participants responded by nominating their telephone number and preferred time to call. All respondents were called and informed of the study’s context. Thirty-three participants were unable to make the available times. A follow-up email was sent to the remaining 50 participants, including an information sheet and a consent form to participate.

### 2.2. Data Collection

All face-to-face focus groups ran on a weekday during business hours, held in university meeting rooms. Two online focus groups, held on weekday evenings, were scheduled through the Zoom platform. Rapport was established, the study and researcher interests and biases were described, and then *a priori* semi-structured questions were asked, including “Can you describe how your feelings and emotions have influenced your eating” and “How do you think food and mood are related?” (see Appendix A for the full list of focus group questions). Discussions continued until all questions were addressed and participants had nothing else to add and data had reached saturation [29]. All focus groups were audio-recorded and transcribed verbatim.

All nine focus groups were facilitated by the lead researcher (ML), a female PhD candidate and psychology tutor. The first focus group was supervised by the primary PhD supervisor as part of qualitative PhD training (Assoc. Prof. SS). The lead researcher jotted field notes directly after the commencement of each focus group and kept a reflexive journal noting thoughts, ideas, feelings, values, concerns, attitudes, beliefs, and any biases throughout the data collection and analysis stages [30].

### 2.3. Data Analysis

Data were explored utilising a six-stage thematic template analysis, as demonstrated in Figure 1 [31]. Stage one focused on transcribing all audio recordings using Otter software (https://otter.ai/, accessed on 10 January 2022), cross-checking for accuracy, deidentifying participant information, and making corrections as necessary. Initial notes were made on first reading. Of the 50 participants invited to review the transcripts, one returned no adjustments, and the remainder declined the invitation.

During the second stage, the lead researcher utilised NVIVO software, version 12 [32] to conduct an iterative and inductive coding process, reading all transcriptions line by line [33]. This stage allowed for utterances to be arranged into nodes—"a collection of references about a specific theme” [32]. In collaboration, two researchers reviewed nodes and patterns in the data and developed codes. During stage three, provisional theme titles were applied to clusters of codes, which were developed using a central concept underpinned by the research questions [34,35]. A final thematic template table was developed, and a consensus was agreed upon across the research team about each theme’s content and scope in stage four (see table in Appendix A). In stage five, the entire data set was reapplied line by line to each theme using NVIVO before the final themes and subthemes were interpreted and reported back to the research team for consensus.

## 3. Results

Three themes were identified during thematic template analysis: (i) *social context: familial and cultural influences in food and mood*, (ii) *social economics: time, finance, and food security*, and (iii) *food nostalgia: unlocking memories that impact mood.* These themes identified the importance of social context, life experiences, beliefs, expectations, and values in the food and mood relationship.

### 3.1. Theme One—Social Context: Familial and Cultural Influences in Food and Mood

In this theme, participants acknowledged their childhood, upbringing, family structures, and food culture as essential parts of the social context surrounding food and the connection to mood. Most of the recollections of family and food highlighted the importance of social context for mood. Mary suggests that her mood may not be impacted by the components in her food but that her mood is more highly influenced by the social context or environment within which she eats:

Maybe it’s not just the food? There’s a whole environment and the context in which you’re in. I think it really makes a difference if you’re in some kind of a family system, or you’re not. I think part of mood, if I’m not being too analytical, is, maybe with feeling ready, feeling prepared, feeling organised. To me, that’s where the comfort level is. Way beyond my own personal tastes for food. Eating with my family makes me happy, despite what foods are being served. Preparing and sharing meals together is a way we express our love and gratitude towards each other.(Mary)

Much of the extract describes the social context of family systems, the preparation and consumption of food, and how this is linked to feelings of comfort and security. A possible consequence of this is that Mary may eat foods that she does not like or does not think are healthy but are preferred /shared by her family. She has a sense of family duty and the role that food plays in this, knowing her family is organised and prepared with food makes her feel positive. She suggests that creating a loving family environment when eating influences her mood positively, and for her, food and mood relationships centre around social environments and context rather than the specific foods or nutrients eaten.

Similarly, Vicki describes that eating a healthy meal in solitude may not provide the psychological benefits of eating in an environment that supports social connection. She describes eating lunch at her work desk and the resultant isolation impacting her mood negatively, compared with the feelings of wellbeing she receives eating lunch outside or with her co-workers:

As far as mood is concerned, I can eat the healthiest diet in the world, in complete isolation. Say if I pack myself this beautiful organic lunch, and it’s got probiotics from the kefir and sauerkraut and all my organic greens, and I go to work in a cubicle, and I don’t talk to anyone. I just sit down at my desk all day and eat lunch at my computer. I just think I’m going to get sick. I don’t think. I’m going to be well. When I eat away from my desk surrounded by others, even if the food itself is not as healthy, I tend to feel more uplifted once I return to my desk.(Vicki)

Vicki suggests that eating healthily while on her own may not benefit her psychologically as when eating in a less healthy manner with others. A potential implication of this is that her work lunchtime routine, eating without stopping or moving away from her workstation, could adversely influence her mood negatively compared with eating in the lunchroom (or away from the workplace) with co-workers. The value Vicki ascribes to eating in isolation suggests that her mood may benefit from eating in a social context with friends or co-workers and may not necessarily come from components of the food she eats.

Another way that family and cultural influences were addressed during the focus group discussions was how current moods manifest by historical eating attitudes linked to childhood, upbringing, and parental eating exemplars. Amy reflects on her mother’s role modelling and the relationship between emotion and food:

Every time I come into a situation that’s highly emotional. I just stuff my face. My Mum doesn’t deal with emotions. If you’re upset, she’ll bring around a cheesecake (laughs) and two spoons. I think a lot of times they don’t know how to comfort and that’s where my Mum brings a cake…... She won’t even give you a hug, so uncomfortable. But she will eat a cake with you. That’s her hug. We know where it’s coming from, a place of love. But it was the way she was raised as well.(Amy)

This extract describes the use of food as a mechanism for emotional expression in a family, where conventional emotional displays were less favoured. Amy suggests a physical and emotional disconnection from her mother, which her mother tries to compensate by using food in times of trouble or sadness. She reflects that this method of using food to show love may have been learned from her grandparents. Amy acknowledges that these learned behaviours passed down from generation to generation are reflected in her adult relationship with food and its link to comfort eating and using food for emotional regulation in her adult life.

Participants reflected on childhood eating and how foods eaten with friends and family increased or decreased mood. The discussions also included descriptions of how eating in a social context with family and friends impacted mood and wellbeing either positively or negatively. Participants discussed the importance of preparing and eating foods in the social environment on mood rather than the nutrients or components in the food providing mental wellbeing. Participants also referred to how families used food to show love towards each other, particularly if this emotion was difficult to express within the family.

### 3.2. Theme Two—Social Economics: Time, Finance, and Food Security

In this theme, participants reflected on how time constraints and other food security issues—such as cost—influenced their food choices and mood. They expressed that organisation, foresight in preparation, and planning helped them make healthier food decisions, which elevated their mood. Others described lack of time, tight finances, feelings of procrastination, lethargy, and decision fatigue as responsible for making less healthier choices, and this lowered their mood. Katherine acknowledges her frustrations with a lack of time to create healthy meals for her family:

I’d make homemade pizzas, we did the base, and we could all put what we liked on them. But that would take me like an hour and a half of making the dough and rolling it out. It’s about time restraints as well, families who work and then pick up kids and then get home, and it’s not just about cost, it’s about the cost of time.(Katherine)

In this extract, Katherine reflects the positive social context of cooking together as a family, but that this was not achievable every night. Time constraints from work and school make it difficult for her family to get together and cook in this way. An implication of this lies in that eating healthily and in a social way that elevates her family’s mood may not always be about the financial cost of the ingredients, but that lack of time and other priorities are also a significant contributor.

Time constraints were a commonly discussed barrier to healthy eating, and having time to prepare food was seen as a luxury to improving mood and health. This acknowledgement of time as a luxury is reflected in Sarah’s description of how food preparation is an important part of her day that helps her limit unhealthy, convenient food options:

I wouldn’t say I’m a food prepper…… sometimes I try, but I find that too boring and too overwhelming. I tend to make lunch at the same time as dinner. If I’m making dinner for my boyfriend. I’m going to make our lunches……… If I’m making this now, I’m just gonna do it double. I always make sure there’s leftovers, put it in the lunch boxes, ready to go. I found that if I didn’t, it would be too hard in the morning. I wouldn’t even be able to stomach the idea of lunch at that time. I would just end up wasting all this money buying stuff out and not feeling that good about it.(Sarah)

Feelings of preparedness were a mediating factor in the food and mood relationship. Sarah reveals high levels of preparation that compensate for her mood. She feels that preparing food in advance helps her control her eating habits. Her mood is elevated by feeling prepared and being in control. Sarah reflects that a lack of preparation would leave her feeling low and lazy, and she fears a lack of control over the types of foods that would end up in her lunch if she was not well prepared. A potential positive consequence of this is that Sarah does not have to waste money buying lunch or being left to deal with food preparation in the morning. Sarah uses the words “overwhelming” and “boring” to describe traditional meal preparation, but she describes how her way of preparing lunches from dinners makes her feel ready and organised for the next day, and this feeling of preparedness elevates her mood.

Participants across the focus group concurred that substantial time, effort, and preparation translated into eating well, but that this investment in time and health delivers rewards in heightened mood levels and greater wellbeing. Mary compares the differences between planning and being prepared and choosing fast, convenient options and their role in her mood:

It takes extra time if you’re going to eat well and prepare well and plan well. I think we all know; if you’re tired or fatigued, it’s too hard to think, you can’t be bothered. You just go and grab whatever, which keeps you in that cycle, and when you’re feeling down, and then you’re tired, you’re like, “I’ll just grab something else.” Now you’re more tired and down. I’ve made all these decisions during the day. I think that’s the hardest part. The whole day like, tick, tick, tick, tick, tick, tick and then you just get to the night and you’re like—well I just had to eat the whole packet of chips.(Mary)

When Mary describes “eating well” in this context, it appears that she means eating healthily. She suggests that, for her, preparation and planning are important aspects of healthy eating. As with Sarah, Mary finds it difficult to be well-planned and prepared when feeling exhausted at the end of the day. She acknowledges the role of decision fatigue at the end of a hard day—after making many healthy decisions throughout the day her ability to make healthy choices in the evening is diminished. Potential implications of her fatigue could result in it being easier to eat an unhealthier, more convenient option, promoting a cycle of unhealthier choices. It is possible that the cognitive dissonance of wanting to eat healthily but instead choosing an unhealthy option lowers her mood and exacerbates her weariness.

In the following extract, Sophie highlights her frustrations with balancing work, study, personal, and family responsibilities and explains that time is an important factor in her and her family’s food choices:

When do you find time to cook? To go shopping, get your whole foods and then go home and cook them and participate in some social activity like soccer? It’s hard, working full time, studying full time, being a parent, and managing time. It’s not so much an emotional thing. It’s more like, I just am spread thin. I’ve got the kids, and I’m like okay, you want frozen pizza for dinner? Hey, let’s just have frozen pizza for dinner. It’s the time thing for me. I would love if I had the energy to always walk into a perfectly clean kitchen. I would love to cook a healthy meal and have healthy snacks around for the kids, me, and everybody, but it’s the time really.(Sophie)

Time constraints negatively impact Sophie’s energy and emotions. She illustrates maintaining a difficult balance between competing responsibilities that detract from her ability to think about, cook, and prepare healthy meals. Sophie discusses how succumbing to unhealthier food choices is sometimes the easiest option due to her time constraints and visualises what a perfect world would look like for her to provide her family with healthy meals, if only she had the time.

Descriptions of social economic stressors for busy, working families included both financial and time pressures. Participants recounted that they did not always provide the healthy, nutritious meals for families as intended to because of these constraints. A common narrative was the influence of decision fatigue. Participants reflected on the many decisions made throughout the day, making it difficult to make healthy choices in the evening, where it is easier to turn to convenient, less healthy choices.

### 3.3. Theme Three—Food Nostalgia: Unlocking Memories That Impact Mood

In this theme, lines of discussion within the focus groups turned towards the role that food and nostalgic memories had in the experience of happy and bittersweet mood. The following extracts explore how certain foods connected participants with fond memories of loved ones who had passed on, creating feelings of nostalgia, and this reconnected them to happy times through the consumption of common foods eaten in their past:

I think there are certain things that are comfort, whether it’s a particular food group, or it’s a memory of a food. If you had something that was wonderful when you were a kid, and you associate food with a wonderful time and you just want to go back there. My Mum always loved jam and cream. You’d have a slice of bread, no butter, jam, and then whipped cream on top, and that was her go-to. I lost her 18 months ago, so I’ve been having this periodically, because it’s what Mum wanted. So that’s the other element of food and mood—because it is memory that connects you to those you love.(Sarah)

Sarah comments on how she can transport herself back to these feelings of her late mother through the food they would eat together. She uses the emotive word “wonderful” when reminiscing about the foods she shared with her mother when she was a child. Sarah recalls her mother’s love for comfort foods and how these foods now instil comfort in her. Food provides her with a connection to memories that inspire a positive mood and help her remember happy times. Similarly, Layla reminisces about food memories from her childhood that bring back nostalgic memories and impact her mood:

Growing up, my mother, who has now passed, cooked desserts. I have strong feelings associated with these foods, and they take me back to my childhood and make me feel comforted. I just need cream. If I have any sort of cake even if it’s cheesecake creamy, I’ll have to have a big glob of cream. This was my mother as well. She was the same.(Layla)

Layla reflects on how she now chooses the same foods she witnessed her late mother eating as a child in an attempt to reconnect with those memories and feelings of comfort and solace. In another discussion, the role of food and nostalgia in mood was addressed by the male participant Jonas. He describes how he developed a love for cooking and preparing food, which ties him to his family who live overseas:

I love cooking. So, I get really excited when I start cooking …. Because my father lives in Greece and I cook there with my Mum and my Dad. It brings me in touch with those moments. So that’s really motivating. I get really happy when I’m cooking. I love having a glass of wine and chuck on the music and there’s a great atmosphere. It’s a big event for me every time I cook, because it makes me feel like I am there with them.(Jonas)

The process of cooking food and being absorbed in the experience of cooking connects Jonas with his family, who all live abroad. The physical act reminds him of cooking with his family and connects him with happy memories. He finds that his family does not need to be directly in the room where he is cooking and eating. Rather, he benefits from the social context through actions, smells, texture, sound, and sights of the food; this connects him with his family when they are not physically present and contributes to elevated mood.

Participants recollected and shared bittersweet memories of their childhood, families, and loved ones both passed and still alive which were inspired by certain types of foods they remember their loved ones eating. Food provided an instant connection and portal to past feelings and emotions which elevated mood. Participants shared that they would consciously seek out specific foods to help with a reconnection to memories of the past and happy times with their loved ones.

## 4. Conclusions

A thematic template analysis of participants’ accounts of food and mood experiences highlighted three themes. In the first theme, *social context: familial and cultural influences in food and mood,* participants expressed that the social context in which they consumed food had a greater influence on their mood than the nutrients in or components of the food itself. They also considered how family circumstances and cultural upbringing influenced the relationship between food and mood. In the second theme, *social economics: time, finance and food security*, participants described how financial and time restraints influenced food choices that subsequently impacted mood. In the third theme, *food nostalgia: unlocking memories that impact mood*, participants suggested that they could reconnect with bittersweet memories from the past by choosing to eat specific foods from their childhood or that loved ones preferred.

The focus group findings highlight the influence of social context and social economics on food choices and mood. Participants suggested that the social context of how food is eaten influenced mood positively and negatively, and past and present familial and cultural factors influenced the relationship between food and mood. It has been proposed that there is a social influence on individuals’ eating behaviours [36]. People’s food choices, eating patterns, and behaviours differ depending on if they are in an environment with other people compared with being alone. Dietary choices change depending on environmental and social context, and these social norms may be viewed as an adaptational response to reward. Social norms influence food choices, and these norms are set by the behaviours of the company at the table and their culture and upbringing [37]. Consequently, social eating norms could play a role in the mood experienced while eating. Current research on social eating suggests that both the quality and quantity of an individual’s social circle are associated with better health outcomes and happiness. Activities that are common during social eating, such as preparing and sharing meals, singing, dancing, laughter, and conversation, lead to increased bonding and wellbeing [38,39,40]. Social eating also triggers endorphins which promote bonding [41]. Research on the social determinants of food choice and diet suggests that mood is a key factor in this relationship [42,43].

Individuals experiencing mental ill-health are more likely to encounter social disconnect, stigma, and loneliness [44,45], which is likely to influence their food choices. Eating with loved ones, family, and friends strengthens social bonds and relationships. Social eating releases neurotransmitters and endorphins that provide bonding and feelings of wellbeing. Social eaters have a greater quality of life, engage more frequently with their local communities, and have a wider circle of close friends, which corresponds with the focus group findings [41]. Other studies suggest that eating alone is associated with lowered mood [46]. Those who ate alone were unhappier than those who regularly ate with others. These studies also suggest that those who experienced mood disorder symptoms and ate alone were more likely to have exacerbated symptoms.

Participants who identified that they noticed a positive mood increase when they prepared meals at home suggested this was due to the deep connection with their family, culture, and the origins of the food they sourced and consumed. The finding within the focus groups of preparing and cooking elevating participants’ mood in the theme *familial and cultural influences in food and mood* was supported by research on the psychosocial benefits of cooking interventions, which found that cooking had positive effects on quality of life, social function, self-esteem, and influence [47].

In the theme *social economics: time, finance, and food security*, participants suggested that food insecurity, time, and financial constraints influenced their food choices, indirectly impacting their mood and mental wellbeing. Food insecurity is defined as the limited availability of nutritional foods and is linked to the stress and uncertainty of not knowing where the next healthy meal comes from [48]. Food insecurity for those classified as lower socioeconomic is directly associated with rising costs and affordability of food and is indirectly influenced by time constraints associated with needing to work longer hours to afford the basic necessities [49]. Furthermore, food insecurity negatively influences mental health and wellbeing. The chronic stress associated with food insecurity can change the biological and psychological reactions to food and how the body metabolises nutrients, leading to worsened physical and psychological outcomes [50]. Those who experience food insecurity are often isolated from their communities and fearful of being embarrassed about their inability to access food and use of food programs. Food insecurity can also lead to feeling demoralised by a lack of food resources and the inability to provide for self and family. Food insecurity also plays a large role in food choices through price, expendable income, food access, and availability [50,51]. Ultimately, the foods that are affordable, easy to attain, and available in food-insecure areas are often processed, convenient or sugary [52], which indirectly impacts the mood and mental wellbeing of people who live in these regions [53]. People in food-insecure areas report experiencing higher levels of depression, anxiety, poor sleep quality, and low mood than people in food-secure areas [54]. This literature and the current focus group findings suggest that food insecurity could negatively influence mood and mental wellbeing, and that this is strongly tied to the unique financial and time constraints of people experiencing food insecurity. Research suggests that individuals with low diet quality, low fruit and vegetable consumption, and high consumption of salty and sugary foods and beverages may be at higher risk of lowered mood and increased depression risk [4,16,55,56].

In the novel and surprising theme *food nostalgia: unlocking memories that impact mood*, participants described a recollection of memories prompted by the consumption of foods from their past or that loved ones’ preferred. Nostalgia is described as a bittersweet recollection of memories and sentimental longing for the past [57]. Food nostalgia is a new and emerging concept with little research and no firm definition [58]. However, an exploratory qualitative analysis of 104 descriptions revealed links to food nostalgia associated with yearning; childhood; homesickness; rediscovery; special occasions; and substitution and that food nostalgia is linked to positive emotions, unlike the traditional definition that nostalgia infers ambivalent or negative emotions [59]. Additionally, Reid, Green, Buchmaier, McSween, Wildschut, and Sedikides [58] describe food nostalgia as a powerful way to evoke psychological benefits from memories that can be elicited from a particular food. Participants in the focus group fondly recollect memories that were directly tied to the consumption of specific foods and how their mood was influenced by being transported back to these past fond memories. Research suggests that individuals use food nostalgia not only to recollect fond memories but as a tool to cope with negative emotions or stress, turning to nostalgic foods as a coping mechanism to counteract negative mood [60]. The results of the current study echo these findings and provide convergent support from a person-centred perspective.

In the current study, participants described social context, food security, and nostalgia as important constructs that impacted their food and mood relationship. These three constructs are tightly coupled: there is a direct link between social connection and nostalgia, and food enhances this connection [61]. There is also a direct link between social connection, nostalgia, and mood, which was identified within the focus group discussions.

### Strengths and Limitations

This study was the first of its kind to explore food and mood in a group of Australian participants. However, it is noted that, due to recruitment through a multicampus Australian university, all participants were well-educated and self-identified as being health conscious during the discussions and when nominating dietary requirements for refreshments. Selection bias may have influenced the discussion, as the voices of those who are less health-conscious were not explored [62]. Additionally, discussions were cross-sectional and relied on memory recall and may be susceptible to recall bias [63]. Social desirability bias may also have impacted the discussions, as participants may have held back any comments they believed did not fit within social norms or did not want to reveal information that contradicted or misaligned with others opinions or insights during the focus group discussions [62]. It would be beneficial for future research to focus on the voices that were not explored in this study by accessing participants who are less health-conscious or providing a semistructured interview structure that could avoid the influence of commentary from other participants.

This study aimed to explore individuals’ experiences with the food and mood relationship by conducting focus groups with Australian participants. Using thematic template analysis, three themes were identified that suggested that social context, food security, and food nostalgia were elements that impacted the food and mood relationship outside the food components or nutrients that are consumed. This research expands on the current quantitative nutritional psychiatry research and suggests that quantitative studies examining the links between diet and mood should look beyond the nutritional factors and give increased attention to the cultural, social, economic, and identity aspects of diet. This type of inquiry could broaden the current field of nutritional psychiatry and expand the context and understanding surrounding the diet and depression relationship.

## Figures and Tables

**Figure 1 ijerph-20-02011-f001:**
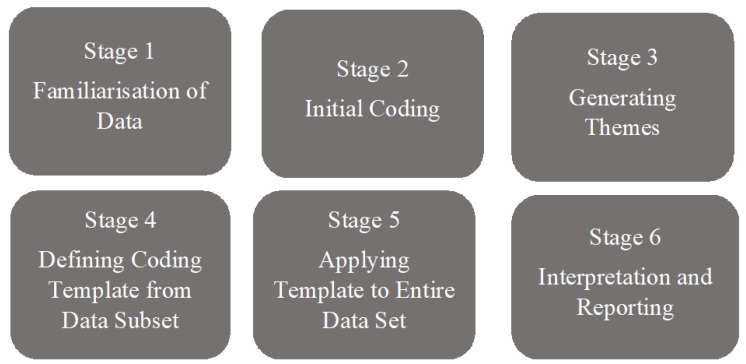
Six Stages of Thematic Template Analysis. *Note.* Reprinted with permission from: The utility of template analysis in qualitative psychology research by J. Brooks, S. McCluskey, E. Turley, & N. King, 2015, *Qualitative Research in Psychology*, 12(2), 202—222 [31]. 2022, Taylor and Francis.

## Data Availability

The transcripts for this focus group project will be made available by a request to the authors.

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
