# Peer review of "“Maybe it’s Not Just the Food?” A Food and Mood Focus Group Study"

_ijerph, 2023, doi:10.3390/ijerph20032011_

Round 1
Reviewer 1 Report
The manuscript “Maybe it’s Not Just the Food?” A Food and Mood Focus 2 Group Study presented to me for review is a very interesting approach to the problem of food intake and the emotions that accompany it.
However, I missed some information at work.
1. What were the exclusion and inclusion criteria.
2. Did all 50 people meet the criteria and therefore be included in the study?
In manuscript Authors wrote that all participants were well-educated and identified as being health conscious. What does it mean?
3. How was health assessed?
4. Has anyone been diagnosed with an eating disorder or depression?
5. Why was the study structured in such a way that gender and age were not taken into account? Does this mean that no statistical significance was observed?
In my opinion, the characteristics of the study group need to be improved.
Thank you
Author Response
Thank you for taking the time to review our manuscript. Please find attached responses to the reviewer comments

Reviewer 2 Report
Reviewer’s results on the article of ijerph-2170832-peer-review-v1
Major comments
This is the article to study an association of food choice with mood, using group discussion among participants (N=50) identified as either holding a current academic qualification or working towards a university degree, aged between 18 and 72. The authors used Zoom platform for group discussion until all questions were answered. They also analyzed six stages shown in Fig.1 for analyzing three themes: one-social context, two-social economics, and three-food nostalgia. As their results, the authors concluded that social context and social economics seemed influential on food choices and mood. My concerns are as the follows:
1. What the authors would draw from the suspected results? In other words, identified these results, what they would want to do for what? To change choices of food or mood?
2. No characteristics of participants were stated. How were age distribution and how were their annual incomes, or business, working hors a week? These characteristics might be influential on their mood or food choice.
As this is seemed so unique for scientific stud mainly in its methodology and questions to answer, when abovementioned questions are answered and if answers are acceptable, this might be accepted.
Minor comments
1. Results might be summarized in tables
2. Characteristics of all participants might be summarized in tables, including mean age, mean income (95% CI), and the other parameters available in public health.
Author Response
Thank you for taking the time to review our manuscript. Please find attached responses to the reviewers comments.
